# The Development of Electroretinographic Oscillatory Potentials in Healthy Young Children

**DOI:** 10.3390/jcm11195967

**Published:** 2022-10-10

**Authors:** Ting Zhang, Jinglin Lu, Zhaoxin Jiang, Li Huang, Jun Zeng, Liming Cao, Xiaoling Luo, Bilin Yu, Xiaoyan Ding

**Affiliations:** State Key Laboratory of Ophthalmology, Retina Division, Zhongshan Ophthalmic Center, Sun Yat-sen University, Guangzhou 510060, China

**Keywords:** amplitude, development, electroretinogram, electroretinographic oscillatory potentials, implicit time

## Abstract

Purpose: This study aimed to summarize the electroretinographic oscillatory potential (OP) responses in healthy young children recorded by RETeval. Methods: By using the RETeval system, we recorded the implicit times and amplitudes of the OPs (OP1-5), in 132 healthy children aged from 0 to 11 years old. The age, gender, and data of implicit time and amplitude of each child were recorded and analyzed. Correlation analysis was performed between age and implicit time/amplitude. Results: No correlation was shown between the implicit times and amplitudes with gender. The implicit times and amplitudes of OP1-5 matured over 10 years of age, with exponential time constants of 1.9, 2.2, 1.8, 1.7, and 1.6 years and 2.1, 2.9, 2.8, 3.0, and 3.2 years, respectively. The majority of the trend occurred within the first 4.6 years. Conclusions: In order to diagnose and evaluate vision-related disorders, the OP response is commonly used. The percentiles and age dependence of OP responses calculated and shown in this study could be regarded as reference data in age-matched pediatric patients.

## 1. Introduction

Oscillatory potentials (OPs), an important part of an electroretinogram (ERG), are evident by superimposing a low-amplitude, high-frequency potentials overlay on the rising phase of the b-wave [1]. Based on pharmacologic manipulations, early OPs have been attributed to activity in bipolar and photoreceptor cells in the outer retina, whereas the later OPs are associated with the inner retinal ganglion and amacrine cells [2,3,4,5,6,7].

Young children with possible retinal diseases may need electrophysiological tests to diagnose them, particularly if their fundus examinations are normal but their visual acuity is significantly diminished [8]. As the most sensitive markers of ERG, OPs indicate significant changes in several retinal diseases at the early stages of the disease, whereas other ERG changes are not as obvious in the fundus [9]. OPs are thought to reflect inner retinal layer function and are sensitive to changes in retinal circulation [7,10]. Moreover, previous studies have reported that OPs are a reliable quantitative method for detecting ischemic retinal diseases in early stages. Therefore, OPs might contribute to the differential diagnosis of retinopathies and facilitate the interpretation of pediatric retinal diseases, such as myopia [11], retinopathy of prematurity (ROP), familial exudative vitreoretinopathy (FEVR), Stickler syndrome, vascular and neuronal diseases, including diabetic retinopathy (DR) [12,13], glaucoma [14], and hypertensive retinopathy, and retinal dystrophies and degenerations, including retinitis pigmentosa (RP), Stargardt’s disease, congenital stationary night blindness (CSNB), X-linked congenital retinoschisis (RS), and Best’s vitelliform dystrophy [15,16,17]. Consequently, the establishment of reference values for OPs in pediatrics is important, as it will help identify increased eye growth in ROP patients prone to developing axial myopia. It is also a good tool for the early recognition of neuronal abnormality and deficient transmission in the Schubert ± Bornschein type of CSNB with an unusual scotopic system [18], and later OPs seem to be involved, as in the early cases of X-linked congenital retinoschisis [19].

However, the OPs from traditional ERG have not been widely used in young children, because of poor cooperation and a need for mydriasis, as well as topical and even general anesthesia. Placing the electrodes of traditional ERG on the conjunctiva or cornea requires specific expertise and a time-consuming examination. Moreover, OPs vary greatly with age in children, and the trend is not yet known. The lack of standardized reference values increases the difficulty of OP applications. As was performed in this study, documenting and analyzing OPs provides a highly sensitive approach for the early recognition of age-related functional changes in the retina of young children. The objective of this study was to establish normative data of OPs in healthy young children (from infancy to 10 years old) to provide a reference for clinical applications.

## 2. Methods

### 2.1. Ethical Guidelines

This was a cross-sectional study conducted between 2017 and 2021, which was performed in accordance with the Declaration of Helsinki. This study was approved by the Institutional Review Board of Zhongshan Ophthalmic Center, Sun Yat-sen University. Because all participants were children, informed consent forms were signed by the parents or guardians.

### 2.2. Subjects

A total of 132 normal subjects, aged 0–11 years, were enrolled in this study. All subjects received comprehensive ophthalmic examinations. The participant populations were all Chinese of Han origin. The inclusion criteria and examination items have been described in a previous paper [20].

### 2.3. RETeval Electroretinographic Oscillatory Potentials

The RETeval system (LKC Technologies, Gaithersburg, MD, USA) was used to record full-field electroretinographic OPs. All subjects were examined without pupil dilation. The device’s specifications have been previously shown [21,22]. Our study involved a mixed rod-cone response and was performed under scotopic conditions. In brief, all patients underwent a 20-min dark adaptation in a dark room with an opaque blindfold on top of an eye patch. The disposable electrodes (sensor strips) were placed 2 mm below the lateral half of the lower eyelid margin after cleansing the skin with a mild dermabrasive gel. After dark adaptation, the RETeval device delivered dim stimuli at intensities of 0.28 Td.s, 85 Td.s, and 280 Td.s in a small Ganzfeld dome with a diameter of 60 mm by intermittent flashing light. The eye position was stabilized by encouraging fixation on a red central fixation point inside the stimulator. Fixation was monitored by viewing the pupil with the RETeval device. After validating ERG waveforms recorded from previous trials and repositioning the mini-Ganzfeld stimulator, trials were performed in duplicate to assess the internal reliability of each stimulation condition.

The reference OPs implicit times and amplitudes were calculated as only the first measurements for each eye. The RETeval system automatically measures and displays the amplitudes and implicit times of the OPs by using special algorithms for discrete Fourier transform and cross-correlation analysis. In general, sedation has been reported to have little effect on the amplitude or waveform of OPs. Chloral hydrate (75 mg/kg) was administered in children with poor cooperation.

### 2.4. Statistical Analysis

Statistical analysis was performed by statistics software (IBM SPSS Statistics 20; IBM Corp., Armonk, NY, USA). Data from both eyes were used according to Davis and Hamilton (2021) [23]. If replicates were available, only the first measurement was used to avoid an undue reduction of device-dependent variability during the construction of the reference intervals.

By using nonlinear curve fitting methods, the age-dependence of the OPs was calculated. The formulas had two terms: a constant term and a decaying exponential which represented the maturation of the eye. Residuals were calculated to assess differences between measurements and the fit. Age dependence was not shown with residuals, so percentiles were calculated from the residuals and added back to the fits to form age-related reference intervals [23].

To determine whether there was a difference between genders, an ANOVA test was conducted. The results were considered statistically significant when *p* was less than 0.05. Analyses were performed with R Statistics. Because the majority of the datasets failed the Shapiro–Wilk normality test, the Mann–Whitney U test and Spearman test were performed to identify group differences.

## 3. Results

### 3.1. Demographic Characteristics of the 132 Healthy Children

The median age of the subjects was 4.7 years old (range 0.3–10.6). The ratio of male to female was 1.16:1. The data of both eyes were used for statistical analysis. The majority of subjects completed the examination of the RETeval assessment successfully. Oral sedation (chloral hydrate 75 mg/kg) was administered in 16 children (16/132, 12.1%) less than 5 years old to complete the comprehensive testing, mainly in boys between 2 and 4 years old.

### 3.2. Comparison of Amplitudes and Implicit Times in OP1-5 by Gender

No statistically significant difference was found in the OP1-5 with gender (Figure 1). The median implicit time of OP1-5 individually, and the *p* values for males and females are shown in Table 1 (Figure 1A). Moreover, the amplitude in males and females are also similar (Table 1, Figure 1B).

### 3.3. Regression Analysis between Implicit Times in OP1-5 and Age

Five non-linear regressions best fit to the data are shown (Figure 2B). The best-fit equations are shown in Figure 2B, where the implicit time is in units of ms, age is in units of years, and e is Euler’s number (~2.71828). Based on this equation, the expected implicit times of the birth of OP1-5 are 18.0, 25.6, 35.1, 43.7, and 48.8 ms. The residuals versus age are shown in Figure 2C. No significant trends were shown in the mean across age, indicating a good fit.

It is apparent in Figure 2B that the implicit times of OP1-5 were more stable in children older than 4.6 years. The ANOVA test showed a negative relationship between the implicit times of OP2-4 and age, with statistical significance (*p* < 0.01). The implicit times significantly decreased with age in children younger than 4.6 years old and slowly decreased further until 11 years of age.

### 3.4. Regression Analysis between Amplitudes in OP1-5 and Age

The best-fit equations for amplitudes are shown in Figure 3B. Age is in units of years, the amplitude is in units of µV, and e is Euler’s number (~2.71828). Based on these equations, the expected amplitudes at the birth of OP1-5 are 1.4, 2.15, 0, and 0.6 µV. The residuals versus age are shown in Figure 3C. There is no significant trend in the mean across age, indicating a good fit.

The ANOVA test showed that the amplitudes of OP1-5 were positively correlated with age, whereas Figure 3B shows that the correlation might mainly exist in children under 4.6 years old. The graph shows that the amplitude increased significantly with growth under 4.6 years old, whereas the trend of increase in the amplitude became smooth in the children who were older than 4.6 years old (Figure 3B).

### 3.5. Suggested Reference Data of Implicit Time and Amplitude of OP1-5 in Children

The scatter in the residuals of both time and amplitude is similar among age groups (Figure 2B and Figure 3B). Therefore, all data were combined to calculate percentiles, which were added back to the age-related fits [15]. The percentiles required for 95% reference intervals are shown in Figure 2. The quantities that need to be added to reach a certain percentile are shown in Table 2.

Representative wave forms of OP1-5 extracted from eight eyes of four subjects (aged 4 months to 10 years old) are shown in Figure 4.

## 4. Discussion

RETeval is a feasible, fast, and effective device for screening young children, which may be an indispensable tool for pediatric ophthalmologists in the future. The OPs of the ERG are a good indicator of early disturbed retinal neuronal function, as well as a compromised microvascular blood flow in retinal disease such as ROP, myopia, and diabetic retinopathy. The OPs, especially those that are scotopically induced, can be used to objectively describe the degree of severity and the progress of the disease. Moreover, a diagnosis may require the use of OPs, especially when the fundus examination is normal and the young child’s vision is significantly limited. However, no publications established reference full-field electroretinographic OP values by using the RETeval handheld device in children. Consequently, this study provides reference scotopic OP data for healthy children by using the RETeval device. The RETeval handheld system was well tolerated by children over 3 years of age, which eliminated the need for sedation in this study. Most younger children were not sedated; only 12.1% (16/132) younger than 4 years were sedated with chloral hydrate to complete the RETeval standard full-field ERG detection (in this study only OPs were analyzed).

OP wavelets have a high frequency and a low amplitude compared to the graded responses of retinal cells represented by a- and b-waves, including OP1-5. In the present study, OP1-3 are distinct from each other, as can be identified in the density plot (Figure 2A and Figure 3A). OP4 and -5 do show some overlap. Whether OPs should be analyzed individually or in groups is not yet conclusive. Moskowitz et al. [8] did not analyze OP1 because it could be contaminated by the a-wave when investigating OPs in 10-week-old infants and comparing them with those in adults. Consequently, they subtracted the rod-mediated a-wave (P3) model from the full ERG to obtain P2 and then filtered that (75–300 Hz) to demonstrate the oscillatory wavelets. The resulting OP1 was much smaller and had a longer implicit time than the OP1 extracted from the full ERG, whereas the amplitude and timing of OP2-5 showed no change [8]. On the contrary, another study on the association between myopia and OPs only revealed mixed OPs. The average peak frequency of rod-driven OP and refractive power was identified and had a significant positive correlation [24]. To our knowledge, although the generation of OPs may be involved in the bipolar, interplexiform, and amacrine cells directly or indirectly, the exact intraretinal sites from which the individual OPs originate remain unknown. In addition, the OPs respectively represent the activation of several retinal generators, whereas different synaptic activities may be the basis of different oscillatory peaks. From this, it can be concluded that using the sum of the amplitude of OPs may limit the possibilities of a more thorough diagnosis and analysis of patients with retinal disease. Certainly because of this, Lachapelle et al. [19] proposed measuring the relative amplitudes of the individual oscillatory peaks. Therefore, in the present study, it seemed prudent to individually analyze OPs instead of merging OP2-4 or all OPs, as has been performed in some studies.

The factors influencing the outcomes of OPs remain controversial, and the association between gender and amplitude is unclear. In the current study, we investigated the effect of gender on OPs with a sample size of more than 100, and there was no dependency of implicit times and amplitudes for OPs from gender. To the best of our knowledge, there are very limited data on OPs according to gender. For instance, Sannita et al. [25] demonstrated that no significant differences between male and female subgroups were observed for any OP parameter, and this is consistent with our findings.

Several authors agree that dark-adapted OPs cannot be recorded in some infants, although some [8] have argued that it is recordable in some preterm infants from 30 weeks’ postmenstrual age, and some [26] have shown it after four weeks of age. Moskowitz et al. [8] concluded that infants’ OPs are remarkably smaller than in adults, with scotopic OPs averaging 19% of that in adults, whereas the amplitudes of the saturated photoreceptor responses are 43% and 66%, respectively. Mean interpeak intervals are similar in infants and adults, indicating oscillatory behavior with a frequency of 155 Hz under scotopic conditions. This is in agreement with the results of Westall et al. [17] and Moskowitz et al. [8], whose research indicated that OPs were the most immature of the ERG responses in early infancy. However, the rate of development has since outpaced the other responses. In addition, when the OPs will stop growing is controversial. Westall et al. [17] reported that OP amplitudes were within adult levels by 21 months. By contrast, Sannita et al. [25] noted an initial increase in oscillatory potential amplitude and a decrease at around age 50 with an inverted U-shaped distribution. However, in our study, it is shown that OPs developed rapidly before the age of four and does not stabilize after the age of four, and the amplitudes were still slowly increasing. Moreover, before the age of four, the growth rate of OP2 was greater than that of other OP amplitudes. However, like OP3 and −4, it entered a period of slow growth after the age of four, rather than an earlier age. Studies of Mactier et al. [27] and Westall et al. [17], which are obviously consistent with our results, respectively, showed that the growth of OP2 amplitude was more rapid than the growth of other OPs. Sannita et al. [25] reported an initial increase in OP amplitudes from childhood to adulthood. Likewise, based on our available data, the amplitude of OPs increases from infancy to 11 years old. Changes in the amplitude of OPs should be considered as part of an age-related baseline when screening for lesions affecting the inner or outer retina, especially in childhood retinal diseases such as ROP, myopia, Stickler syndrome, and FEVR. From a practical perspective, when the diagnosis of these diseases is considered in young children by ERG testing, the age of children must be considered to distinguish whether the prolonged implicit time is due to physiological or pathological reasons.

Therefore, the most important finding to emerge from the analysis is to the precise determination of how implicit time and amplitude change with age. This study found that timing and amplitude maturation, respectively, are well-fit by exponential functions. Reference intervals can be calculated for any age (less than 11 years old) with these formulae.

There are several limitations in our study. First, the extracted OP waveforms were affected deeply by filter types and their parameters, including the creation of artifactual components. It suggested that a better definition of the filtering technique in ERG procedures and standard so that we are in a better position to predict the outcome from an OP point of view. To be in a better position to predict the outcome from an OP point of view, a better definition of the filtering technique in ERG procedures and standard were necessary. When the filter used is the same as that used in this study, the normative data can act as a reference. Secondly, the skin electrodes used by the RETeval system are designed for adults and older children, so it may be difficult to place them at appropriate marks on a baby’s face. Therefore, the electrodes may not fully adhere to the skin under the eyelids. Therefore, the amplitude levels may be underestimated. Thirdly, the pupillary area was recorded but not analyzed in this study. Although Kato et al. found the pupillary area to be an independent factor of implicit timing in flicker ERG [11], in the same study, they found that the RETeval system delivers a stimulus with unchanging retinal illuminance when the pupil diameter was less than 6.3 mm (equal to the pupillary area of 33.2 mm^2^). As the natural pupillary diameter is usually less than 6.5 mm, they suggested that this mydriasis-free system is valid.

Overall, regression analyses of RETeval OP results in 132 healthy children showed that age is an independent factor for OPs, which affects both the implicit times of OP2-4 and the amplitudes of OP1-5. Age should be carefully considered when analyzing OPs. The medians and ranges obtained in 132 healthy children could serve as reference data against which the OP responses from age-matched pediatric patients can be compared. These data will help provide electrophysiology labs with information for assessments of clinical pediatric ERGs.

## Figures and Tables

**Figure 1 jcm-11-05967-f001:**
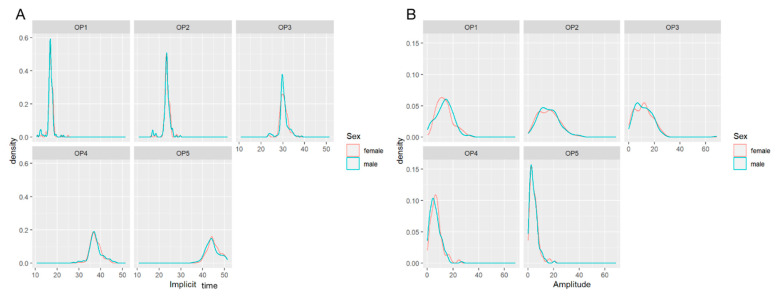
Relationship between implicit times (**A**)/amplitudes (**B**) and gender of healthy children. There is no significant correlation between gender and the implicit times/amplitudes of the OP1-5.

**Figure 2 jcm-11-05967-f002:**
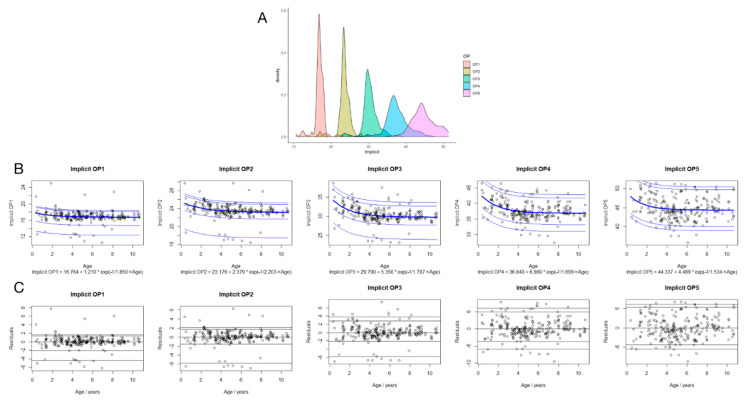
The implicit times of OPs dependence on age. The density plot for the implicit times of OP1-5 (**A**). OPs implicit times (**B**) are shown versus age. The fit equation for the bold curve (labeled 50th percentile) is shown at the bottom of the plot. Other percentiles with additive offsets are shown with thin blue lines. Fit residuals of regression ERG dependence on age (**C**). Fit residuals or differences between measurements and fitted equations are shown as implicit times. Each patient has one symbol per eye.

**Figure 3 jcm-11-05967-f003:**
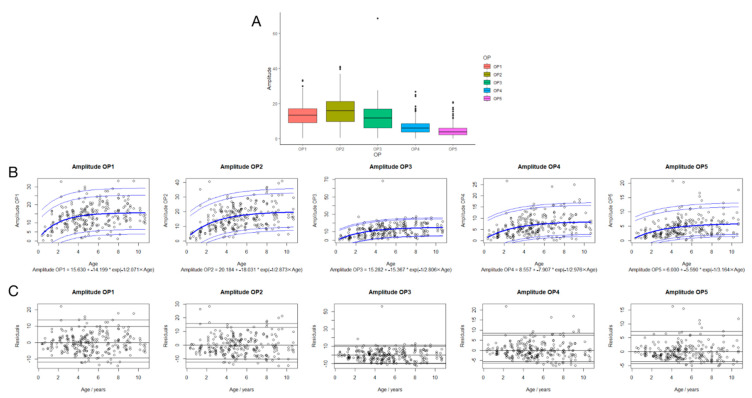
The amplitudes of OPs dependence on age. The density plot for the amplitudes of OP1-5 (**A**). OPs amplitudes (**B**) are shown versus age. The fitted equation for the bold curve (labeled as the 50th percentile) is shown at the bottom of the plot. Other percentiles with additional offsets are shown with thin blue lines. Fit residuals of regression ERG dependence on age (**C**). Fit residuals, or the difference between the measured value and the fitted equation, are displayed as amplitudes. Each patient has one symbol per eye.

**Figure 4 jcm-11-05967-f004:**
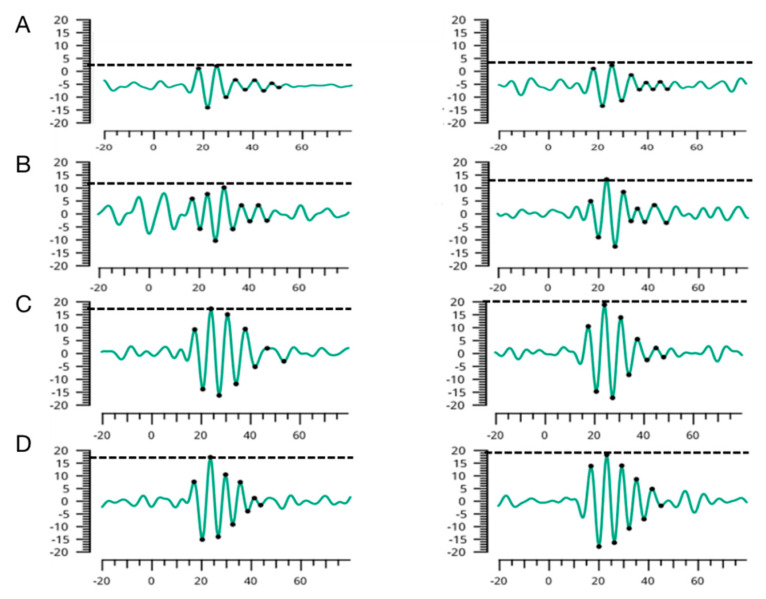
Representative OPs in children of different ages. Representative OPS of both eyes are shown for a 4-month_x0002_old infant (**A**), and 5-year-old (**B**), 7-year-old (**C**), and 10-year-old (**D**) healthy children. Dashed lines indicate the highest peaks.

**Table 1 jcm-11-05967-t001:** The median implicit times of OP1-5 individually and *p*-values for males and females.

		Median, Range		
		Male	Female	*p*
Implicit time (ms)				
	OP1	16.9 (11.0–22.9)	17.0 (10.6–25.0)	0.180
	OP2	23.5 (16.8–29.7)	23.6 (16.4–29.7)	0.185
	OP3	30.1 (23.3–38.6)	30.5 (23.2–37.5)	0.625
	OP4	37.3 (27.4–46.5)	37.3 (29.6–44.2)	0.963
	OP5	44.0 (35.8–51.2)	44.5 (37.8–51.4)	0.083
Amplitude (μV)				
	OP1	13.9 (0.3–33.1)	12.0 (0.4–33.2)	0.975
	OP2	15.0 (0.5–40.5)	16.0 (2.0–41.0)	0.803
	OP3	11.5 (0.4–27.0)	11.5 (0.2–27.5)	0.727
	OP4	3.7 (0.2–20.4)	3.9 (0.1–20.8)	0.100
	OP5	5.3 (0.1–26.7)	6.3 (0.3–24.2)	0.318

**Table 2 jcm-11-05967-t002:** Additive amounts for 2.5%, 5%, 95%, and 97.5%.

	Implicit Times (in ms)				Amplitudes (in µV)			
	2.5%	5.0%	95.0%	97.5%	2.5%	5.0%	95.0%	97.5%
OP1	−4.38	−2.02	1.37	1.63	−11.79	−9.20	9.76	13.64
OP2	−5.76	−1.56	1.73	2.12	−12.47	−10.25	12.99	16.02
OP3	−5.78	−2.07	2.83	3.79	−9.92	−9.35	9.92	11.30
OP4	−5.93	−3.70	4.87	5.95	−6.42	−5.53	7.48	8.63
OP5	−5.42	−4.31	5.31	6.06	−4.31	−3.59	5.93	7.32

## Data Availability

The authors declare that all data supporting the findings of this study are available upon request to the corresponding author.

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
