# Peer review of "The Development of Electroretinographic Oscillatory Potentials in Healthy Young Children"

_jcm, 2022, doi:10.3390/jcm11195967_

Round 1

Reviewer 1 Report

This paper reports on the developmental profile of the oscillatory potentials of the electroretinogram in Chinese children in Guangzhou up to the age of 11. The methods and the result seem solid, but there is a major problem with their interpretation. The paper purports to report normative data. But normative data in the presence of a high level of disease or disorder is a difficult concept, because agreement with the norm may not indicate lack of disease and departure from the norm may not be a good marker of the development of a disease.

It is well-known that the prevalence of myopia is very high in Chinese children, with the prevalence of myopia reaching approximately 50% by the end of primary school. Since myopia is a disorder that results from cumulative excessive axial elongation, signs of abnormality may appear earlier than the onset of myopia. It is not clear if the word “healthy” includes low likelihood of subsequent development of myopia, but I doubt it. The authors need to clearly define what they mean by healthy, and if this term covers the majority of Chinese children, who develop myopia by the end of schooling. If this is the case, then the data describe norms, in the sense that these are normal results from the population, but they do not describe “healthy norms.”  There is a similar problem in relation to blood pressure. Initially, normal blood pressure was defined in terms of what was normal within the population, and they tended to rise, until they started to be defined in terms of what produced healthy long-term outcomes, and then acceptable blood pressure by age started to drop. Confusions of this kind can have significant health consequences.

The other conceptual issue concerns curve fitting. The authors fit the data with exponential functions, and argue that this means they can confidently interpolate up to the age of 11. Depending on the density of the data on which the curve fitting is based, this seems a reasonable approach. However, the authors follow this up with claim that, given the form of the equation, it will also be useful for adults. Unless there is a theoretical reason for expecting the type of curve obtained, extrapolation is far more problematic, and since I suspect that there is no theoretical basis for the curve, the authors should argue that whether it applies to adults requires more collection of data on older subjects to establish the shape of the curve.

There are a number of minor grammatical errors in the paper, and the authors should seek help from a native English speaker to correct them.

The references also need attention. For example, reference 8 is cited as Anne et al, but Anne is the first name of the first author. This is the reverse of the common mistake with Chinese names made of westerners.

Author Response

 Thank you very much for your thoughtfulness. The inclusion criteria and exclusion criteria have been elaborated in our previous study. The inclusion criteria included the following: (1) all infants and children born at term(40 ± 2 weeks) gestation, (2) best-corrected visual acuity (BCVA) of at least 20/25 Snellen visual acuity in children older than 3 years of age, no requirement for BCVA in the children younger than 3 years old,(3) IOP ≤20 mm Hg, (4) optic nerve cupping <50%, and (5) refractive error (spherical equivalent) between −3.0 D and +3.0 D. Those who had any known ocular or systemic diseases or myopia of -3.0 diopters (D) or more were excluded. If a subject satisfies all of the above conditions, it will be considered a “healthy” subject.

In this paper, this sentence, “based on the form of the equation, we expect that it will also be a useful model for adult subjects as well”, was mentioned. However, what we wanted to express is that this form of the equation may be used to research ERG in adult subjects, not mean that the equation summarized from healthy children will also be useful for adults. We just advanced a research method. Because of this sentence caused misunderstanding, we deleted it from the paper.

We agree with you so much, according to your suggestions, we corrected our grammatical errors in the paper.

Reviewer 2 Report

This is an important manuscript that provides baseline reference normative data for the RETeval handheld ERG device. I thoroughly enjoyed reading it. I do have some thoughts and suggestions.

Can you provide some additional details regarding the race characteristics of the participant population. Although age has been considered as a factor that can influence oscillatory potentials, it has been reported that race may influence ERG results. This is a factor that needs to be considered when describing normative data. 

Some additional detail regarding the number of children in each age group (perhaps in 6 monthly blocks until the age of 4 years and then yearly after the age of 4) would be useful for interpreting the data. For example, when looking at the graphs of the dependence of OP amplitudes on age, it appears that are much fewer children under the age of 1 compared to children aged 3 to 4 years. If there are a very small number of children in particular categories, then a single large outlier could potentially impact the predictions significantly. 

I would like some more details regarding the number of children where chloral hydrate was necessary to perform the examination. Was there a specific criteria for this like a fixed number of failed attempts before proceeding? Was there a failure rate for obtaining the examination at all?

I also noted that duplicates were performed for the purposes of within reliability of each stimulus. However no results were presented with regards to this? How reliable was the test on repeats. Have you calculated intraclass correlation coefficients for this?

Just a minor writing point, when referencing Moskovitz et al in the discussion, the authors are using the author's first name "Anne". I believe normally it should be the author's surname that's used. 

Author Response

Thank you very much for your insightful suggestions. The participant populations were all Chinese of Han ethnic origin.

The large outliers that were more than 3 standard deviations from the mean were excluded so that they will not impact the predictions.

There was a specific criterion for children where chloral hydrate was necessary. If the gaze was at the indicated fixation position for less than 1 second and failed attempts arrived three times, children <4 years old received chloral hydrate, and children>4 years old were removed. The failure rate for obtaining the examination was 5% approximately.

Duplicates were performed. If data of multiple measurements were close, the first measurement data were used. Otherwise, they were removed.

We thank you so much. According to your suggestions, we corrected our grammatical errors in the paper.